# The effect of chronic ankle instability on muscle activations in lower extremities

**Chiao-I Lin**[1,2]*, **Mina Khajooei**[1], **Tilman Engel**[1], **Alexandra Nair**[1,2], **Mika Heikkila**[1], **Hannes Kaplick**[1], **Frank Mayer**[1]

1 Outpatient Clinic, University of Potsdam, Potsdam, Germany, 2 Department of Physical Activity and Health, Sociology of Health and Physical Activity, University of Potsdam, Potsdam, Germany

* chialin@uni-potsdam.de

## Abstract

### Background/Purpose

Muscular reflex responses of the lower extremities to sudden gait disturbances are related to postural stability and injury risk. Chronic ankle instability (CAI) has shown to affect activities related to the distal leg muscles while walking. Its effects on proximal muscle activities of the leg, both for the injured- (IN) and uninjured-side (NON), remain unclear. Therefore, the aim was to compare the difference of the motor control strategy in ipsilateral and contralateral proximal joints while unperturbed walking and perturbed walking between individuals with CAI and matched controls.

### Materials and methods

In a cross-sectional study, 13 participants with unilateral CAI and 13 controls (CON) walked on a split-belt treadmill with and without random left- and right-sided perturbations. EMG amplitudes of muscles at lower extremities were analyzed 200 ms after perturbations, 200 ms before, and 100 ms after (Post100) heel contact while walking. Onset latencies were analyzed at heel contacts and after perturbations. Statistical significance was set at $alpha \leq 0.05$ and 95% confidence intervals were applied to determine group differences. Cohen's $d$ effect sizes were calculated to evaluate the extent of differences.

### Results

Participants with CAI showed increased EMG amplitudes for NON-rectus abdominus at Post100 and shorter latencies for IN-gluteus maximus after heel contact compared to CON ($p<0.05$). Overall, leg muscles (rectus femoris, biceps femoris, and gluteus medius) activated earlier and less bilaterally ($d = 0.30–0.88$) and trunk muscles (bilateral rectus abdominus and NON-erector spinae) activated earlier and more for the CAI group than CON group ($d = 0.33–1.09$).

**Data Availability Statement:** All relevant data are within the manuscript and its Supporting information files (S1 File).

**Funding:** The authors received no specific funding for this work.

**Competing interests:** The authors have declared that no competing interest exist.

## Conclusion

Unilateral CAI alters the pattern of the motor control strategy around proximal joints bilaterally. Neuromuscular training for the muscles, which alters motor control strategy because of CAI, could be taken into consideration when planning rehabilitation for CAI.

## Introduction

Ankle sprain is a common injury among athletes [1]. After the first acute ankle sprain, 40% of people will develop chronic ankle instability (CAI) which causes giving way, ankle instability or recurrent ankle sprain [2, 3]. CAI can lower the quality of daily life, affect functional performance, and is related to posttraumatic osteoarthritis [4]. An updated model of CAI suggests that damaged proprioceptors of sprained ankle ligaments can alter signal inputs to the central nervous system and consequently influence neuromuscular control [4]. In addition, this change affects not only the neuromuscular control of the ankle joint but also that of the proximal joints [4].

Altered neuromuscular control of the lower extremities has been shown in the individuals with CAI [4]. Individuals with CAI have shown an inverted and dorsiflexed ankle before heel contact together with a lateral shift of the center of pressure and greater ground reaction force, which may relate to recurrent ankle sprains and giving way [5]. Knees and hips have been shown to compensate for the altered ankle kinematics [6]. Individuals with CAI displayed an earlier and greater vastus medialis activity while walking and less knee flexion and increased hip flexion while landing than the control group [7, 8]. However, the evidence of the effect of CAI on muscle activity at the knee and hip is controversial [8, 9]. Moreover, the evidence regarding alterations to the trunk is scarce [5].

Because of the affected sensorimotor system, CAI affects not only the injured ankle but also the uninjured ankle [4, 10]. Individuals with unilateral CAI showed impaired proprioception, neuromuscular and postural control in both the injured and uninjured ankle [10]. Individuals with unilateral CAI with an affected uninjured ankle could potentially increase the risk of a recurrent ankle sprain on the injured side [11]. The uninjured ankle shows decreased compensatory postural adjustments [11]. Therefore, when the injured ankle encounters an accidental sudden ankle inversion, the uninjured ankle cannot reduce the potential further injury to the injured ankle because it cannot efficiently decelerate the movement of the center of mass in the medial and lateral direction for the injured ankle [11].

Proximal joints play an important role related to ankle injuries [12]. The lumbo-pelvic hip complex (core muscles) is important in controlling hip adduction and in regulating hip internal rotation. Hip adduction affects the postural stability of the frontal plane, and hip internal rotation affects muscle activations on lower extremities and ankle posture [13, 14]. Low core stability and hip muscle strength are risk factors for injuries in the lower extremity and for ankle sprain in athletes [12]. However, the effect of unilateral CAI on neuromuscular control of proximal joints bilaterally is unclear.

Methods of walking perturbation have been applied to investigate dynamic postural control [15, 16]. The perturbations can provoke a muscular reflex, a compensation for an unexpected perturbation, at the lower extremities and trunk [16]. The ability to return to postural stability following external unexpected perturbations is important because it is the performance of core stability and relates to the risk of lower limb injuries [12]. Besides, at heel contact during walking, the core muscles activate because the impact force of the heel strike is conducted from the

lower extremities through the hip to the trunk [12]. At the same time, initial of heel contact is also the timing of weight acceptance from unloading to loading, when ankle sprain commonly occurs [17, 18]. Thus, observing the motor control strategy of proximal joints during walking and after an external perturbation in individuals with CAI could give an insight into the sensorimotor deficiency in CAI.

Therefore, the purpose of this study was to compare the difference of the motor control strategy in ipsilateral and contralateral proximal joints while unperturbed walking and perturbed walking between individuals with CAI and matched controls. To answer this question, the magnitude and onset of muscle activation around the proximal joints during unperturbed and perturbed walking trials on the injured and non-injured side were compared in individuals with unilateral CAI to a group of matched control participants. It was hypothesized that individuals with CAI show altered patterns of muscle activation on proximal joints bilaterally compared to controls.

## Materials and methods

The study design was cross-sectional. The sample size was calculated (Gpower software; Effect size $d$: 1.16; α level: 0.05; power (1-β error probability): 0.80) based on the results of a previous study [19]. All the measurements were conducted in the biomechanics laboratory on the campus of the university from June to December 2019.

## Participants

Thirteen individuals with unilateral CAI in the CAI group (25.2±4.5 years, 168.1±7.9 cm, 65.4 ±9.5 kg) and 13 individuals in the control group (CON) (26.8±4.0 years, 168.6±7.7 cm, 63.3 ±7.6 kg) matched by age, sex, height, weight and dominant leg participated in this study. Participants were physically active (more than 120 minutes of exercise per week) and over 18 years old. The dominant leg was defined as the leg used to kick a ball [20]. All participants were recruited by convenience sampling using flyers on the university campus and sports clubs from May to December 2019.

Participants in CON had no previous ankle injuries. The CAI group met the criteria recommended by the International Ankle Consortium [3]: (1) a history of at least one significant lateral ankle sprain which occurred at least 12 months prior to enrollment in this study. (2) History of 'giving way' on the previously injured ankle, which occurred at least twice in the past six months, and/or recurrent sprain, which happened twice or more in the injured ankle, and/or 'feeling of ankle instability' in their daily or sports activity. Feeling of ankle instability was evaluated using the Cumberland Ankle Instability Tool (CAIT) [21].

Participants were excluded when they (1) had a history of previous surgeries or a fracture of the musculoskeletal structures in either lower extremity; or (2) had acute musculoskeletal injuries of lower extremities within the previous three months; or (3) had CAI and were attending regular balance training; or (4) were not able to complete the questionnaires: CAIT, and Foot and Ankle Ability Measure (FAAM) [3, 21, 22]; or (5) had other neurological disease affecting the sensorimotor system; or (6) were ankle sprain copers; or (7) had bilateral CAI. All participants read and signed the informed consent form. This study was approved by the local ethical committee of the university (43/2019).

## Test procedure

Participants went through a physical examination, preparation for surface electromyography (EMG), maximum voluntary isometric contraction (MVIC) tests, and two walking trials.

## Physical examination

Demographic data were collected, and participants filled out CAIT and FAAM to determine their level of ankle instability and ankle function [3, 22]. Ankle instability was defined as the score of CAIT≤24 [3, 21]. The impaired function was defined as a score of less than 90% in the activities of the daily living subscale of FAAM (FAAM_ADLs) and less than 80% in the sport subscale of FAAM (FAAM_SPORT) respectively [3]. A physician then evaluated the participants to determine the presence of a mechanical ankle instability using ankle anterior drawer test and tilt test [18].

## EMG

Participants' skin was shaved, abraded, and cleaned with a disinfectant before placing bipolar surface electromyography electrodes (2 cm inter-electrode distance, pre-gelled (Ag/AgCl), type P-00-S, Ambu, Medicotest, Denmark). A wireless EMG capture system (band-pass filter: 5–500 Hz, gain: 5.0, overall gain: 2500, sampling frequency: 4000 Hz, RFTD-32, myon AG, Switzerland) was used to record signals with internal signal processing (bandwidth 5–500 Hz, Butterworth filter 4th order, digitized). EMG electrodes were placed on the skin above the rectus femoris (RF), biceps femoris (BF), gluteus maximus (Gmax), gluteus medius (Gmed), rectus abdominus (RA) and erector spinae (ES) bilaterally, according to SENIAM guidelines [23].

## MVIC test

Participants were instructed to perform sub-maximal familiarization sessions as a warm-up before the actual tests to confirm adequate performance. Each test included three sets of five seconds of MVIC knee extension/flexion (RF/BF) [24], hip extension/abduction (Gmax/Gmed) [25], and trunk flexion/extension (RA /ES), during which EMG signals of target muscles were recorded (Fig 1).

## Walking trials

There were two walking trials: an unperturbed walking trail and a perturbed walking trail. In the unperturbed walking trial, participants walked for a duration of five minutes at a velocity of 1 m/s. In the perturbed walking trial, participants walked for a duration of six minutes at the same velocity, during which 10 right- and 10 left-sided perturbations for a total of 20 perturbations were applied in a randomized order 200 ms after initial right heel contacts. The perturbations of the right side were triggered by a load cell embedded in the treadmill on the right belt (customized software solution: Stimuli, pfitec biomedical systems, Germany). Perturbations of the left side were also triggered by right side heel contact and were programmed to occur 200 ms plus stride duration (averaged from 3 stride durations of the unperturbed walking) which was measured during unperturbed walking using a 3D-motion capture system (Vicon MX3, 8 cameras, 200 Hz, Vicon, Oxford, UK). During perturbations, one of the treadmill belts decelerated to a velocity of -1 m/s (amplitude: 2 m/s; deceleration = $-40m/s^2$) in 50ms, after which the belt returned to baseline velocity in 50ms. To detect heel contacts and the onsets of perturbation, the acceleration sensors (ACC) were placed on participants' shoes. The signals from ACC were synchronized with EMG.

## Data processing

EMG data were analyzed using a lab view based analysis tool (IMAGO process master pfitec, biomedical systems, Germany). The highest values (root mean squares, RMS) from three MVIC tests were applied to normalize the EMG data to each recorded muscle. To determine

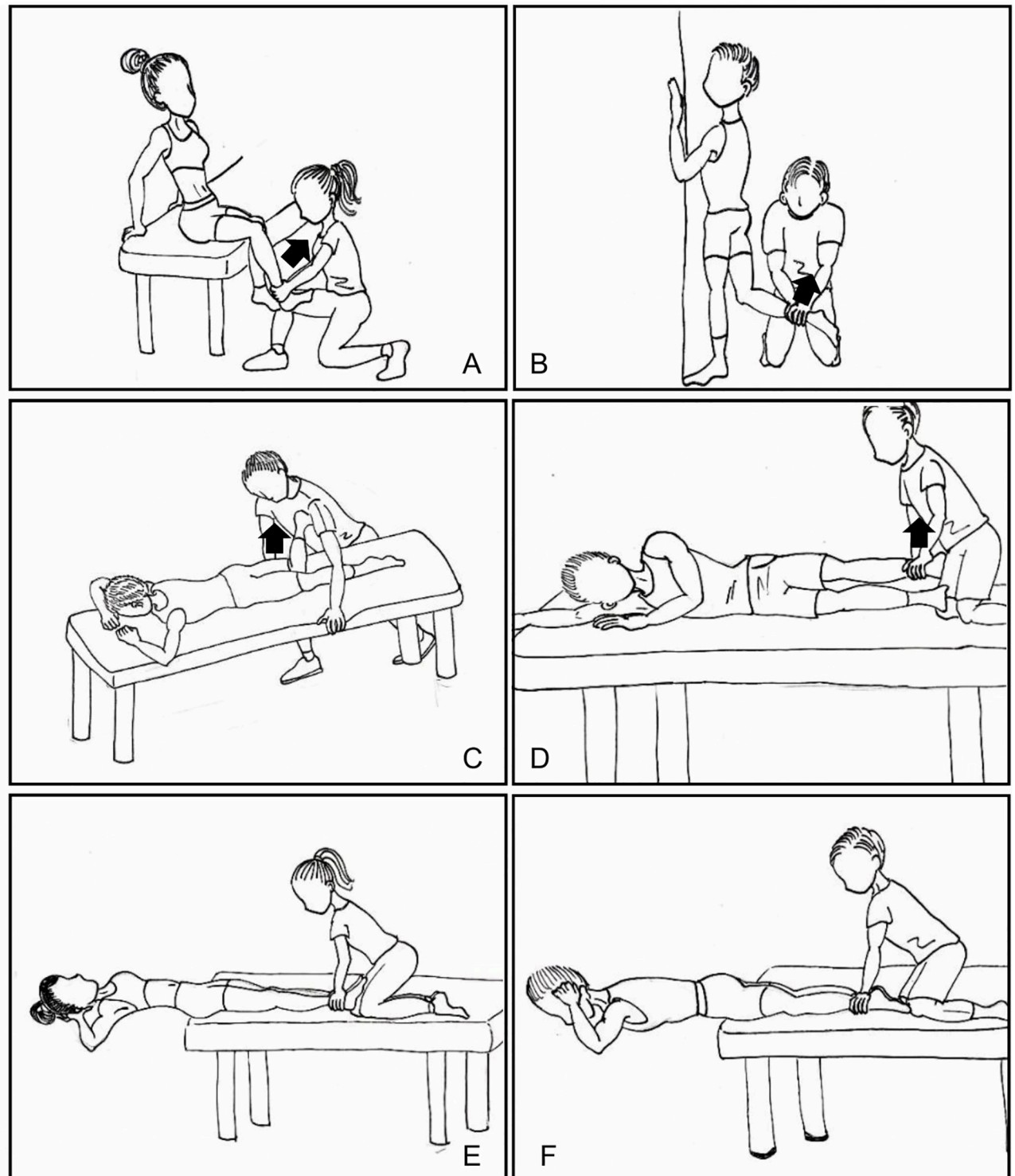

**Fig 1. Maximum Voluntary Isometric Contraction (MVIC) tests for six targeting muscles.** A: Rectus femoris (knee extension). B: Biceps femoris (knee flexion). C: Gluteus maximus (hip extension). D: Gluteus medius (hip abduction). E: Rectus abdominus (trunk flexion). F: Erector spinae (trunk extension).

the magnitude of muscle activation, normalized RMS signals of each muscle were analyzed at three time points: 200 to 0 ms before initial heel contact (Pre200), 0 to 100 ms after initial heel contact (Post100) in the unperturbed walking trial, and 0 to 200 ms after the perturbed (Pert200) in the perturbed walking trail. Clear characteristics of acceleration, detected from ACC, allowed to find the precise timing of heel contact and onset of perturbation when analyzing EMG data (See Figs 2 and 3).

RMS from 10 strides from the walking trial, 10 left-perturbed strides, and 10 right-perturbed strides from the perturbed walking trial were averaged and normalized to RMS of MVIC (%). All EMG signals were smoothened, rectified and filtered as RMS (using 2$^{nd}$ order, low pass 15 Hz) [8, 16].

To determine the onset of muscle activation (ms), an automated detection method was used. The timing when RMS exceeded two standard deviations from baseline was detected by an analysis software (IMAGO process master pfitec, biomedical system, Germany) and was defined as the onset of heel contact during the unperturbed walking trial and at the beginning of perturbations during the perturbed walking trial. In cases where the automatic detection algorithm failed to detect the onset or the onset was considered invalid because of artifacts, the data analyzer would correct it manually [16].

## Statistical analysis

For demographics, CAIT, and FAAM, descriptive statistics with mean and standard deviation were calculated and tested for between groups differences using independent samples t-tests or Mann-Whitney U tests, depending on the distributions. The Shapiro–Wilk test was applied to examine the normal distribution of the data. To detect the group difference in EMG amplitude and onset at different measuring time points, Independent $t$-tests or Mann-Whitney U tests were applied. The level of statistical significance was set at $p \leq 0.05$. Significance levels were not adjusted for multiple comparisons based on Hopkins's and colleagues' recommendations [26]. Additionally, 95% confidence intervals (95%CIs) were also calculated. If 95% CIs did not cross zero, then it was considered a group difference. Cohen's $d$ effect size was calculated to show the magnitude of difference [8, 9, 19, 27, 28]. A small, medium, and large effect size were defined as $\geq 0.49$, 0.50–0.79 and $\geq 0.80$ [29]. All statistical analyses were conducted using SPSS Statistics 25 (Chicago, IL, USA).

# Results

Age, height, weight and weekly hours of physical activity showed no group differences ($p > 0.05$), but the CAI group had a higher level of ankle dysfunction and instability than CON based on the score of FAAM_ADLs (CAI: 89.5±10.5, CON: 98.2±3.0; $p < 0.05$), FAAM_SPORT (CAI: 82.7±13.1, CON: 98.1±3.7; $p < 0.05$) and CAIT of the injured ankle (CAI: 20.7±4.3 and CON: 29.1±1.6; $p < 0.05$).

## Walking trail

Results of the magnitude and onset of muscle activation during the unperturbed walking trail are summarized in Table 1.

**The magnitude of muscle activation.** Generally, the CAI group activated the hip and knee muscles less and the trunk muscle more than CON.

At Pre200 both sides of RF, BF and Gmed activated less; and RA of both sides and the ES of the non-injured side (NON-ES) activated more in the CAI group than CON with a small to median effect sizes (0.30–0.79).

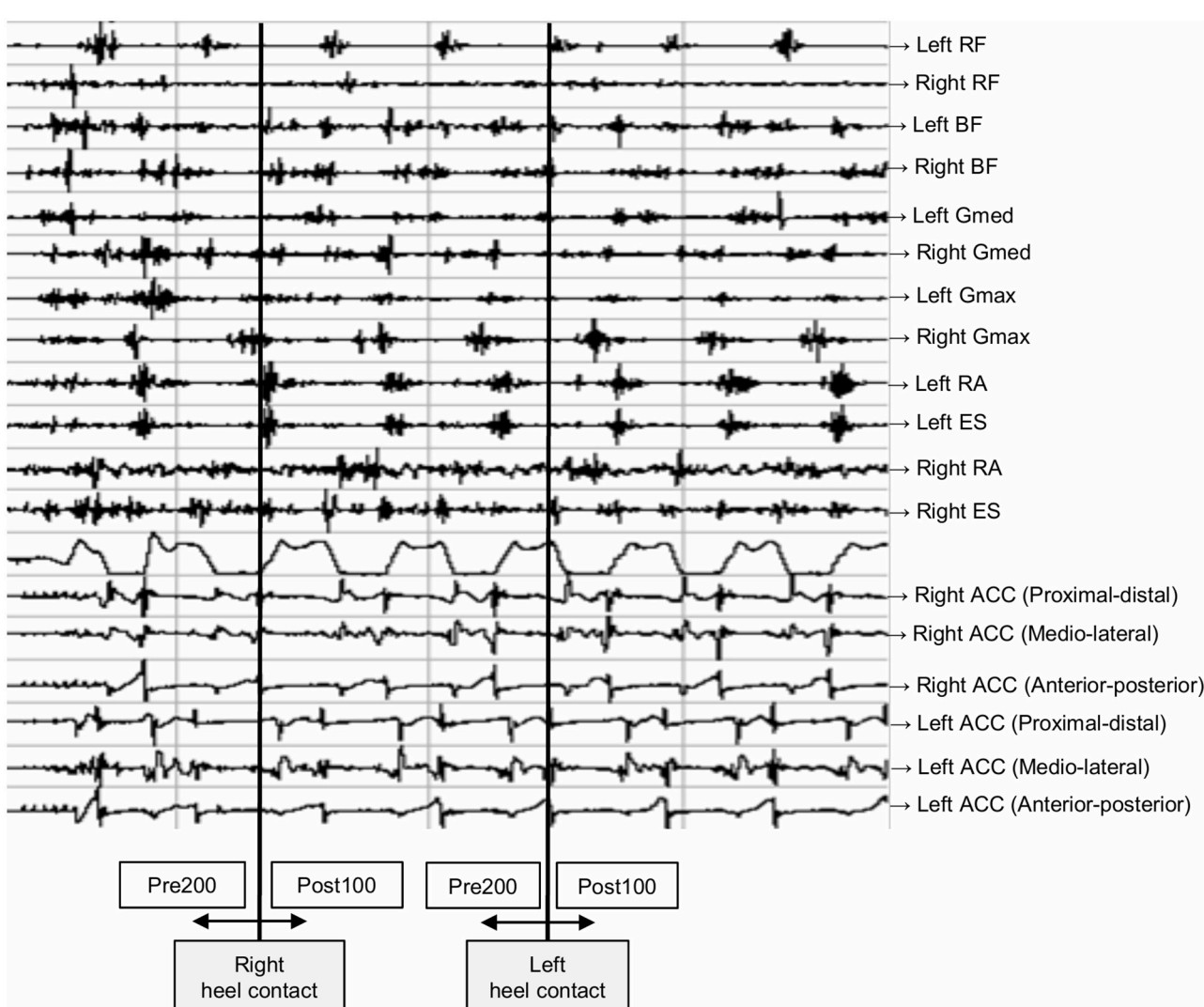

**Fig 2. Visualized signals for each recorded muscle and the acceleration sensors in the electromyography analysis software during an unperturbed waling trial.** Heel contacts provoked visible spikes, which assisted the data analyzer in detecting the heel contacts. Pre200: 200 to 0 ms before initial heel contact. Post100: 0 to 100 ms after initial heel contact. RF: rectus femoris, BF: biceps femoris, Gmed: gluteus medius, Gmax: gluteus maximus, RA: rectus abdominus, ES: erector spinae. ACC: the acceleration sensors.

At Post100, RF, Gmed and BF of both sides and BF of the non-injured side (NON-BF) activated less; and RA of both sides activated more in the CAI group than CON with small to large effect sizes (0.34–0.88).

**The onset of muscle activation.**   After heel contact, the CAI group activated BF of the injured side (IN-BF), Gmax of the injured side (IN-Gmax), and Gmed of the non-injured side (NON-Gmed) earlier than CON with small to large effect sizes (0.43–1.09). The onset of RA was not analyzed after heel contact, because the muscle activity was not clearly detectable for most of the participants (21 out of 26 and 15 out of 26 were undetectable on RA of the injured side (IN-RA) and NON-RA).

## Perturbed walking trial

Results of the magnitude and onset of muscle activation during perturbed walking are summarized in Table 2.

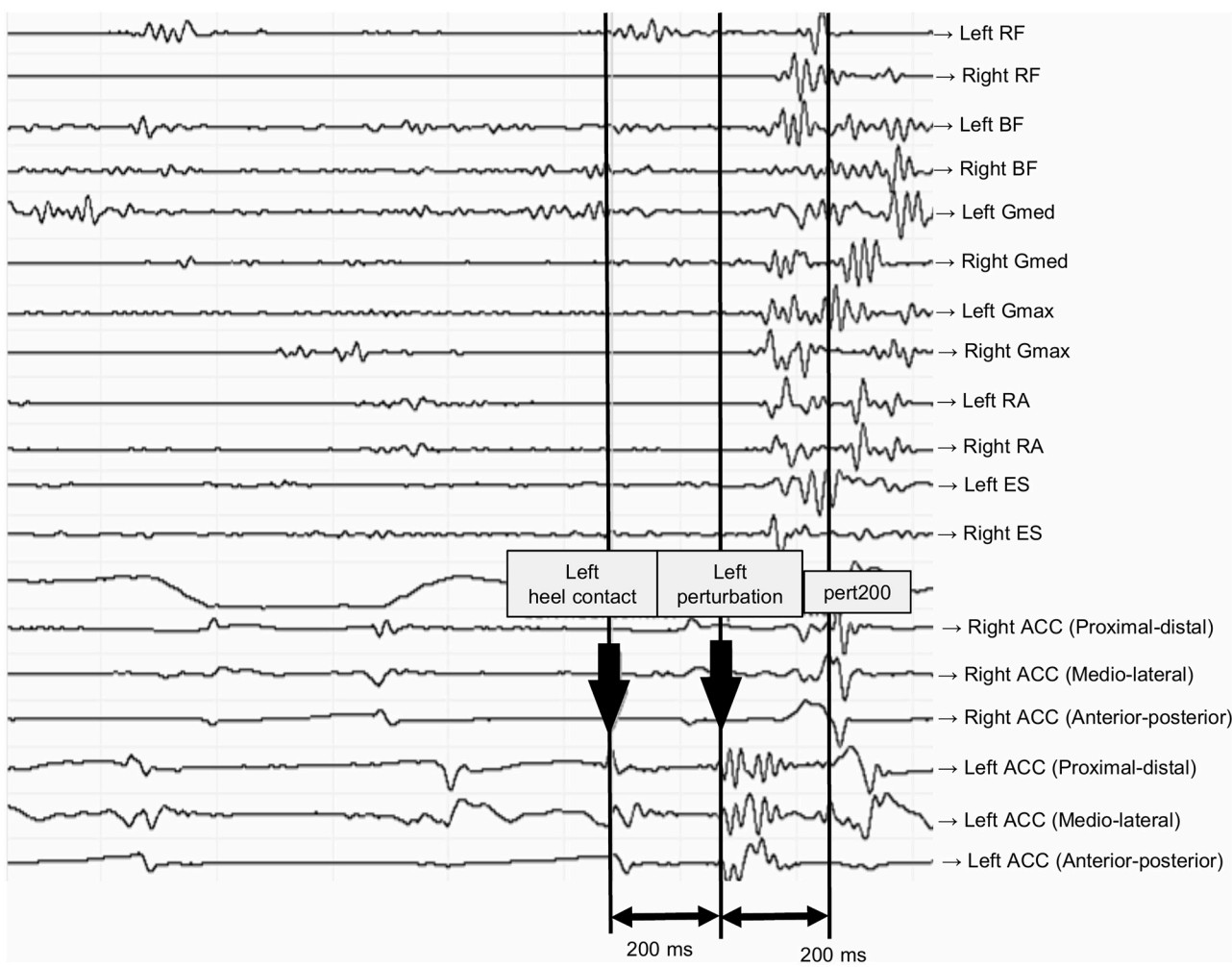

**Fig 3. Visualized signals for each recorded muscle and the acceleration sensors in the electromyography analysis software during a left-walking perturbation.** Onset of perturbation provoked visible spikes, which assisted the data analyzer in detecting onsets of perturbation. RF: rectus femoris, BF: biceps femoris, Gmed: gluteus medius, Gmax: gluteus maximus, RA: rectus abdominus, ES: erector spinae. ACC: the acceleration sensor. Pert200: 0 to 200 ms after the perturbation in the perturbed walking trail.

**The magnitude of muscle activation.** Generally, RF, BF and Gmed activated less; and Gmax, ES and RA activated more in the CAI than CON. At Pert200, BF and Gmed of both sides, and the injured side of BF (IN-BF) activated less in the CAI group than CON with small to medium effect sizes (0.38–0.66). RA of both sides, Gmax of the injured side (IN-Gmax) and NON-ES activated more in the CAI than CON with small to medium effect sizes (0.43–0.54).

**The onset of muscle activation.** After perturbation, IN-Gmax, ES of the injured side (IN-ES), and NON-RA activated earlier, and NON-ES activated later in CAI group than CON with small to medium effect sizes (0.33–0.66).

## Discussion

Individuals with CAI altered the pattern of muscle activation around proximal joints during unperturbed and perturbed walking in accordance with the initial hypothesis of this study.

Gmed, overall, had a lower activation in the CAI group than controls. In the non-injured side, the CAI group activated Gmed earlier during heel contact than CON. The current

**Table 1. Group means, *Cohen's d* effect size, and group difference for the magnitude and onset of muscle activation in the unperturbed walking trail.**

| Muscles | | magnitude (%) | | | | | | | |
|---|---|---|---|---|---|---|---|---|---|
| | | Injured side | | | | Non-injured side | | | |
| | | CON | CAI | p | *Cohen's d* (95%CI) | CON | CAI | p | *Cohen's d* (95%CI) |
| Pre200 | RF | 6.8±4.3 | 5.6±3.3 | 0.63 | **0.30(0.00, 0.61)**§ | 12.8±26.3 | 05.0±2.2 | 0.52 | **0.40(0.10, 0.71)**§ |
| | BF | 19.9±12.5 | 11.8±6.3 | 0.06 | **0.79(0.46, 1.11)**# | 16.4±9.4 | 13.1±8.0 | 0.37 | **0.36(0.06, 0.67)**§ |
| | Gmed | 10.6±5.9 | 6.8±3.0 | 0.10 | **0.76(0.44, 1.08)**# | 9.7±5.2 | 7.4±3.3 | 0.21 | **0.51(0.20, 0.82)**# |
| | Gmax | 21.6±17.7 | 17.4±13.0 | 0.63 | 0.26(-0.05, 0.56) | 14.5±6.56 | 16.6±15.9 | 0.49 | -0.17(-0.47, 0.13) |
| | RA | 5.1±2.6 | 31.2±82.3 | 0.08 | **-0.43(-0.74, -0.12)** | 4.2±2.5 | 29.9±77.6 | 0.06 | **-0.45(-0.76, -0.14)**§ |
| | ES | 14.3±6.7 | 13.0±6.3 | 0.77 | 0.24(-0.09, 0.57) | 14.0±5.7 | 31.8±57.2 | 0.56 | **-0.42(-0.75, -0.08)**§ |
| Post100 | RF | 17.7±14.8 | 11.0±4.6 | 0.59 | **0.58(0.27, 0.90)**# | 23.5±39.0 | 9.0±3.1 | 0.25 | **0.50(0.19, 0.82)**# |
| | BF | 8.7±4.6 | 9.5±5.9 | 0.71 | -0.15(-0.45, 0.16) | 12.2±9.2 | 9.4±6.4 | 0.40 | **0.34(0.03, 0.64)**§ |
| | Gmed | 42.8±18.3 | 28.6±12.1 | 0.06 | **0.88(0.55, 1.21)**✳ | 39.8±15.6 | 29.7±20.3 | 0.09 | **0.53(0.22, 0.85)**# |
| | Gmax | 52.6±34.4 | 52.1±24.4 | 0.66 | 0.02(-0.28, 0.32) | 50.8±36.0 | 47.7±32.5 | 0.78 | 0.09(-0.22, 0.39) |
| | RA | 5.4±3.2 | 36.1±100.8 | 0.30 | **-0.41 (-0.72, -0.11)**§ | 3.81±2.60 | 43.1±123.9 | **0.03***  | **-0.43(-0.74, -0.12)**§ |
| | ES | 13.0±5.9 | 14.4±4.4 | 0.68 | -0.17(-0.50, 0.16) | 13.0±7.0 | 30.7±55.2 | 0.53 | -0.40(-0.74, 0.07) |
| | | Onset (ms) | | | | | | | |
| Heel contact | RF | 71.4±65.8 | 85.6±30.9 | 0.51 | -0.28(-0.61, 0.05) | 73.4±72.4 | 80.7±36.0 | 0.98 | -0.13(-0.43, 0.17) |
| | BF | 189.8±53.3 | 169.5±39.8 | 0.53 | **0.43(0.10, 0.77)**§ | 178.1±31.7 | 172.8±38.7 | 0.78 | 0.15(-0.15, 0.45) |
| | Gmed | 59.8±35.0 | 47.79±109.7 | 0.70 | 0.15(-0.18, 0.48) | 50.5±49.5 | 25.30±61.6 | 0.22 | **0.45(0.14, 0.76)**§ |
| | Gmax | 95.7±49.0 | 41.07±51.5 | **0.01***  | **1.09(0.71, 1.46)**✳ | 76.1±50.5 | 65.62±36.4 | 0.71 | 0.24(-0.07, 0.54) |
| | RA | NA | NA | NA | NA | NA | NA | NA | NA |
| | ES | -5.2±235.4 | -85.5±266.1 | 0.22 | 0.32(-0.04, 0.68) | -90.3±492.8 | -72.3±247.1 | 0.92 | -0.05(-0.35, 0.26) |

RF: rectus femoris, BF: biceps femoris, Gmed: gluteus medius, Gmax: gluteus maximus, RA: rectus abdominus, ES: erector spinae, Pre200: 200 to 0 ms before initial heel contact. Post100: 0 to 100 ms after initial heel contact. CON: the matched control group, CAI: the group of chronic ankle instability, 95%CI: 95% of confident interval for Cohen's d,

* representing p < .05,

✳ representing 95%CI did not cross zero with a large effect size,

# representing 95%CI did not cross zero with a median effect size,

§ representing 95%CI did not cross zero with a small effect size

findings are consistent with two previous studies showing Gmed activating less in the CAI group during the stance phase of walking [9, 28]. DeJong and colleagues used ultrasound imaging and Son and colleagues applied surface EMG to investigate the influence of CAI on Gmed during walking. Both studies applied confidence intervals as a measure to determine the group difference by 90% and 95% CIs respectively [9, 28]. However, not all previous studies are in agreement with the current results [8, 9, 19, 28, 30]. Two previous studies found no group difference [8, 27], and one found that Gmed activated more in the CAI group than CON [19]. This might be caused by different methods used in the studies.

The decreased Gmed activation in individuals with CAI may be an indicator of an altered motor control [4]. As shown in a previous investigation, to compensate for the ankle instability, individuals with CAI activated the peroneal longus earlier and to a higher degree [4]. Rio and colleagues found that participants without CAI showed an ankle-dominant strategy to maintain postural stability, but individuals with CAI shifted from an ankle- to a hip-dominant strategy to maintain balance [6, 9]. However, based on Moisan et al. and Son et al. both suggested that the effort of an altered dominant strategy was insufficient [5, 9]. Because of the decreased Gmed activation, the hip being more adducted results in a lateral shift of the center of mass [9]. Studies showed that in the participants with CAI their center of mass still

**Table 2. Group means, *Cohen's d* effect size, and group difference for the magnitude and onset of muscle activation in the perturbed walking trail.**

| | | magnitude (%) | | | | | | | |
|---|---|---|---|---|---|---|---|---|---|
| | Muscles | Injured side | | | | Non-injured side | | | |
| | | CON | CAI | p | Cohen's d (95%CI) | CON | CAI | p | Cohen's d (95%CI) |
| Pert200 | RF | 90.8±67.7 | 68.2±45.7 | 0.37 | **0.38 (0.07, 0.68)**[§] | 102.3±88.3 | 65.2±31.5 | 0.36 | **0.54 (0.23, 0.85)**[#] |
| | BF | 64.0±93.3 | 34.2±23.2 | 0.23 | **0.42(0.11, 0.73)**[§] | 40.0±14.2 | 34.6±26.1 | 0.19 | 0.25(-0.06, 0.55) |
| | Gmed | 58.6±28.5 | 46.4±22.9 | 0.22 | **0.45(0.14, 0.76)**[§] | 56.8±24.4 | 40.9±21.9 | 0.11 | **0.66(0.34, 0.97)**[#] |
| | Gmax | 58.4±48.3 | 82.9±48.9 | 0.07 | **-0.48(-0.79, -0.17)**[§] | 68.3±42.6 | 56.05±51.6 | 0.29 | 0.25(-0.06, 0.55) |
| | RA | 29.5±21.9 | 40.7±43.2 | 0.34 | **-0.43(-0.74, -0.13)**[§] | 29.1±31.4 | 125.88±289.2 | 0.17 | **-0.47(-0.78, -0,16)**[§] |
| | ES | 91.3±67.0 | 78.5±28.6 | 0.80 | 0.26(-0.07. 0.59) | 91.0±46.1 | 106.17±30.3 | 0.20 | **-0.54(-0.88, -0.20)**[#] |
| | | Onset (ms) | | | | | | | |
| perturbation | RF | 101.1±18.7 | 98.0±14.6 | 0.64 | 0.18(-0.15, 0.51) | 97.6±20.1 | 93.9±15.3 | 0.69 | 0.21(-0.12, 0.54) |
| | BF | 61.1±48.9 | 56.1±17.6 | 0.77 | 0.14(-0.19, 0.46) | 68.0±30.2 | 66.5±17.0 | 0.45 | 0.06(-0.27, 0.39) |
| | Gmed | 119.9±23.1 | 118.3±31.9 | 0.56 | 0.06(-0.27, 0.38) | 131.4±33.5 | 124.1±35.8 | 0.56 | 0.21(-0.12, 0.54) |
| | Gmax | 140.5±41.3 | 85.6±109.7 | 0.07 | **0.66(0.32, 1.01)**[#] | 117.6±38.1 | 127.3±49.1 | 0.64 | -0.22(-0.55, 0.11) |
| | RA | 115.1±17.5 | 109.3±44.9 | 0.18 | 0.17(-0.16, 0.50) | 114.1±20.0 | 105.9±9.2 | 0.30 | **0.42 (0.08, 0.75)**[§] |
| | ES | 81.7±16.5 | 74.2±27.7 | 0.64 | **0.33(0.00, 0.66)**[§] | 81.8±17.1 | 86.6±9.4 | 0.39 | **-0.36 (-0.69, -0.02)**[§] |

RF: rectus femoris, BF: biceps femoris, Gmed: gluteus medius, Gmax: gluteus maximus, RA: rectus abdominus, ES: erector spinae, Pert200: 0 to 200 ms after the perturbed in the perturbed walking trail. CON: the matched control group, CAI: the group of chronic ankle instability, 95%CI: 95% of confident interval for Cohen's d,

[*] representing p < .05,

[✻] representing 95%CI did not cross zero with a large effect size,

[#] representing 95%CI did not cross zero with a median effect size,

[§] representing 95%CI did not cross zero with a small effect size

excessively shifted laterally, which puts the ankle in a vulnerable position and lead to a potential risk for recurrent ankle sprain in individuals with CAI [5, 9, 14]

In the current study, IN-Gmax activated significantly earlier after heel contact in the CAI group than CON, and the 95% CIs showed that IN-Gmax activated earlier and to a higher degree after the walking perturbation. To the best of the authors' knowledge, muscle activities in individuals with CAI have not been investigated using a perturbation with the anterior-posterior direction. Previous evidence showed that individuals with CAI activated Gmax more than CON in a landing and ball kicking task [4], which is consistent with the current study. No group difference has been found in the magnitude of muscle activation at Pre200 and Post100, which is consistent with the works done by Koshino et al. and DeJong et al. [28, 30].

The results of this study indicate that CAI altered the capability of compensation for unexpected external perturbations. Chuter and de Jonge suggested that core stability maintained by increased hip stiffness is in response to equilibrium perturbation, to force conduction to the spine and force absorption [12]. Another possible explanation could be that, based on the works from Rio et al. and Son et al., individuals with CAI showed a hip-dominant strategy (increased Gmax activity), which is a sign of sensorimotor deficits [6, 31]. Unlike Gmed, which assists in maintaining postural control in the frontal plane, Gmax and knee muscles maintain postural stability in the sagittal plane while walking and can attenuate impact by eccentric contraction [32]. However, previous studies showed that individuals with CAI presenting increased vertical ground reaction forces while walking is an indicator of a reduced capability of impact absorption [4, 5, 33].

The current results showed that the CAI group had smaller RF activation than CON based on the 95% CIs. This result is consistent with Son's and colleagues' work, which found individuals with CAI had less muscle activation of vastus lateralis while walking [9]. However,

Koshino and colleagues found no difference in RF activity between groups [27], and other studies noted that individuals with CAI activated RF 108 ms earlier [8] and 51% higher [33] during heel contact than in CON. In the current study, the 95% CIs revealed that BF activated less at Pre200 bilaterally, IN-BF activated less and later after heel contact and NON-BF activated less at Pert200 in the CAI group than CON. In contrast, Feger et al. and Koshino et al. found no difference of BF activation in a walking task between groups [8, 27].

RF also assists with impact absorption [8, 33]. Hamstrings play an important role in the deceleration of knee extension at the end of a swing phase by eccentric contraction and in acceleration of the leg at initial heel contact by concentric contraction [34]. Reduced RF and BF activation while walking may affect the efficiency of impact absorption, which in turn may change the articular cartilage catabolism and is related to the cartilage degeneration [4, 9].

The varying results could be explained by different methods of RMS normalization and walking speed. Previous studies used static standing [19, 35], peak RMS during walking [33], or static squats [9] to normalize RMS, but the current study and Koshino's and colleagues' work [27] applied MVIC to normalize RMS. Individuals with CAI had altered motor strategy to activate muscle during functional tasks causing reduced or increased muscle activity [36]. This makes the results difficult to compare. Furthermore, walking speeds were also different among studies (self-selected speed or speed ranged from 1.1 m/s to 2.5 m/s), which may also cause the discrepant results [8, 9, 27]. Slower walking speed attenuates muscle activity of the lower extremities [37]. Although the current study applied slower walking speed (1.0 m/s) than the previous studies, the results are similiar to three previous studies which utilized faster walking speed (1.1 m/s, self-selected speed, and 1.35 m/s) [9, 28, 33]. This may imply that walking speed might not affect the presence of muscle activity in CAI.

The current results showed that at Post100, NON-RA activated significantly more in the CAI group than CON (p = 0.03), and 95% CIs revealed that bilateral RA and NON-ES activated higher in the CAI group than CON. After the perturbation in the perturbed walking trial, NON-RA and IN-ES activated earlier, and NON-ES activated later in CAI group more than CON. The connection between individuals with CAI and trunk stability has been investigated [38, 39]. Individuals with CAI showed delayed reflex activity of ES and RA after a sudden unload of trunk perturbation during flexion and extension movements [38], and had less hemidiaphragm contractility than CON [39]. Decreased hemidiaphragm contractility may reduce the intraabdominal pressure which is related to the trunk and postural stability [39]. However, the current results must be interpreted with caution because the inter-subject variability was high.

The increased and early activation in trunk muscle could be explained by a study performed by Nadler and colleagues, which supports the notion that athletes with overuse injuries or ligamentous laxity on lower extremities tend to develop low back pain [40]. The injured joints altered the force of transfer in the lower extremities and decreased shock absorption, which is transferred to the spine and can cause potential low back pain development [40]. The increased and early activation in the trunk muscles is related to a guarding mechanism to protect the lower back [41] or a result of limited ankle dorsiflexion in individuals with CAI which increases the range of RA activation [5, 42].

There were some limitations to the current study. First, it is not clear whether the neuromuscular deficit was a pre-existing condition or the results of CAI. Second, it is not clear if the neuromuscular strategy is affected differently by different stages of CAI development. A participant with a longer history of CAI may have more obvious signs of neuromuscular deficiency than a participant with a shorter history. A longitudinal study should be conducted to confirm it.

## Conclusion

Altered muscle activations around proximal joints in individuals with CAI were found in the current study, which indicates that the motor control strategy of participants with CAI was changed. The results support the notion that CAI affects motor control on proximal joints bilaterally. The changes may relate to the development of further injuries. These findings could be taken into consideration by clinicians when planning rehabilitation programs for CAI. Training the muscles around proximal joints bilaterally should be incorporated when managing individuals with CAI.

## Supporting information

**S1 File. Root mean squares of six measured muscles at Pre200, 100 ms, and Pert200 ms and onsets of muscle activations at heel contact and perturbation during unperturbed walking trail and perturbed walking trail.** Averaged root mean squares of the corresponding time frame and onsets of muscle activation from ten unperturbed walking gaits and ten perturbations from each participant were presented in the dataset.
(XLSX)

## Acknowledgments

We want to express our gratitude to Doctor Kim Loose for performing physical examinations, to Anika Schönefeld for managing measurement appointments, to the whole staff of the outpatient clinic of University of Potsdam for supporting this study, to Anne Puschmann and Michael Fliesser for helping to recruit participants, and to Arash Asefi Rad and Myounghwee Kim for assisting with measurements. We are thankful to Henry Robert Mumm and Sanne Houtenbos for their help with proofreading.

## Author Contributions

**Conceptualization:** Chiao-I Lin, Mina Khajooei.

**Data curation:** Chiao-I Lin.

**Formal analysis:** Chiao-I Lin.

**Investigation:** Chiao-I Lin, Mina Khajooei, Tilman Engel, Alexandra Nair, Mika Heikkila.

**Methodology:** Chiao-I Lin, Mina Khajooei, Tilman Engel, Hannes Kaplick.

**Project administration:** Chiao-I Lin, Mina Khajooei.

**Software:** Hannes Kaplick.

**Supervision:** Frank Mayer.

**Validation:** Chiao-I Lin, Mina Khajooei, Tilman Engel, Hannes Kaplick.

**Writing – original draft:** Chiao-I Lin.

**Writing – review & editing:** Chiao-I Lin, Mina Khajooei, Tilman Engel, Alexandra Nair, Mika Heikkila, Hannes Kaplick, Frank Mayer.

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
