## [Decision Letter · Decision Letter 0]

3 Dec 2020

PONE-D-20-26118

The Effect of Chronic Ankle Instability on muscle activations in lower extremities

PLOS ONE

Dear Dr. Lin,

Thank you for submitting your manuscript to PLOS ONE. After careful consideration, we feel that it has merit but does not fully meet PLOS ONE’s publication criteria as it currently stands. Therefore, we invite you to submit a revised version of the manuscript that addresses the points raised during the review process.

We look forward to receiving your revised manuscript.

Kind regards,

Yumeng Li

Academic Editor

PLOS ONE

Journal Requirements:

2.) Please include additional information regarding the survey or questionnaire used in the study and ensure that you have provided sufficient details that others could replicate the analyses. For instance, if you developed a questionnaire as part of this study and it is not under a copyright more restrictive than CC-BY, please include a copy, in both the original language and English, as Supporting Information.

3.) We note that you have indicated that data from this study are available upon request. PLOS only allows data to be available upon request if there are legal or ethical restrictions on sharing data publicly. For information on unacceptable data access restrictions, please see http://journals.plos.org/plosone/s/data-availability#loc-unacceptable-data-access-restrictions

Additional Editor Comments:

Some major revisions are needed based on the reviewer's comments.

Reviewers' comments:

Reviewer's Responses to Questions

**Comments to the Author**

1. Is the manuscript technically sound, and do the data support the conclusions?

Reviewer #1: Partly

2. Has the statistical analysis been performed appropriately and rigorously? 

Reviewer #1: Yes

3. Have the authors made all data underlying the findings in their manuscript fully available?

Reviewer #1: No

4. Is the manuscript presented in an intelligible fashion and written in standard English?

Reviewer #1: Yes

5. Review Comments to the Author

Reviewer #1: Summary:

Individuals with chronic ankle instability and a control group walked on a split-belt treadmill while perturbations were introduced. Electromyography from muscles of the trunk and thigh were evaluated 200ms before, 100 ms after, and 200 ms after the perturbation to assess response amplitude and latencies.

Comments:

The authors have presented data to indicate that neuromuscular differences are present between individuals with CAI and healthy matched controls. The premise of this manuscript is commendable. There are areas where the authors need to further detail their procedures and/or explain themselves with more clarity. I would like to suggest that the authors also correct sentence structures and grammatical errors in the document.

Lines 97-99: “Therefore, the purpose of this study was to investigate the effect of unilateral CAI on the ipsilateral and contralateral muscle activations around proximal joints while walking.” Actually, based upon the background and ideas the authors have presented, the purpose of their study is to observe the effects of an external perturbation on the neuromotor control of proximal muscles in individuals with and without unilateral CAI. In doing so, they can compare differences between these groups regarding the motor control strategies.

Lines: 131-132: what questionnaires did the participants complete?

Lines 166-170: a graphic would assist the reader in determining how MVIC were performed or the authors can specify how the participants were positioned while performing the MVIC. It is also not clear what value from the MVIC was used to normalize the EMG

After reading the Data Processing within the Methods section, the authors stated that they compared average EMG profiles from 10 strides in each of the walking conditions. It is not clear how EMG and the perturbation were synchronized in order to accomplish this. The authors stated that a load cell was instrumented on the treadmill. Were these synchronized so that heel contact could be determined?

In the Results section, it is suggested that the sections titled “The magnitude of muscle activation “ and “The onset of muscle activation” be compartmentalized into analyses based upon the conditions (walking and perturbation). This will provide better clarity to the reader and better organization on the authors’ part. Similarly, tables 1 and 2 can be clarified further to match the text.

The authors indicated that other studies required participants to perform walking trials at 1.3-1.5 m/s, while the participants in this study walked at 1.0 m/s. It is not clear why this discrepancy was present, and this could influence the interpretation of the data with the literature.

Minor comments:

Lines 53-55: neuromuscular of ankle joint? Do the authors intend to state the neuromuscular control of the ankle joint?

Lines 72-75: this sentence needs further clarification. Do the authors intend to state that the uninjured ankle is incapable of movement control that would reduce the potential for further injury to the already injured ankle?

Lines 87-89: “The compensation of external unexpected perturbations” – are the authors suggesting that compensatory mechanisms are essential to maintain system stability when external perturbations are presented? If so, this is not how their sentence is communicated

Lines 175-179: were these 10 total perturbations or 10 on the left and 10 on the right for a total of 20?

Lines 180-184: was this same process used in the right side perturbation?

Lines 271-273: it is not clear what the authors are stating. Gmed had lower activation overall, but activated earlier at heel contact in the CAI group compared to controls?

Lines 288-291: the shift of the center of mass that the authors bring forward is unclear. During the gait cycle the center of mass shifts laterally depending upon the limb supporting the body. Do the authors indicate that excessive lateral shift occurs in individuals with CAI?

Lines 315-317: if there is no statistically significant difference, then there is no difference. Other areas of the Discussion section are stated similarly, and need to be adjusted accordingly (ex, lines 321-324). This section needs to be rewritten to reassess the authors’ findings with the current literature

Tables 1 and 2: are the statistics provided the comparisons between the healthy group and the CAI group? This is unclear in the legends

6. PLOS authors have the option to publish the peer review history of their article (what does this mean?). If published, this will include your full peer review and any attached files.

Reviewer #1: No

---

## [Author Response · Author response to Decision Letter 0]

17 Dec 2020

Dr. Yumeng Li and the reviewer,

Thank you for the suggestions. I wrote responses point by point.

Additional requirements from the PLOG ONE: 

Response: Yes, checked.

2) Please include additional information regarding the survey or questionnaire used in the study and ensure that you have provided sufficient details that others could replicate the analyses

Response: Citations of the two questionnaires applied in the current study (the Foot and Ankle Ability Measure and the Cumberland Ankle Instability Tool) were added in the manuscript. The original articles provide the full version of the questionnaires. 

3) We note that you have indicated that data from this study are available upon request. PLOS only allows data to be available upon request if there are legal or ethical restrictions on sharing data publicly.

Response: All relevant data are within the manuscript and its Supporting Information files (S1 file).

b) If there are no restrictions, please upload the minimal anonymized data set necessary to replicate your study findings as either Supporting Information files or to a stable, public repository and provide us with the relevant URLs, DOIs, or accession numbers. 

Response: The minimal anonymized data set has been uploaded.

From the reviewer:

The responses to each point: 

1. To correct sentence structures and grammatical errors in the document.

Response: A colleague did proofreading for this manuscript (Line 449). 

2. Lines 97-99: “Therefore, the purpose of this study was to investigate the effect of unilateral CAI on the ipsilateral and contralateral muscle activations around proximal joints while walking.” Actually, based upon the background and ideas the authors have presented, the purpose of their study is to observe the effects of an external perturbation on the neuromotor control of proximal muscles in individuals with and without unilateral CAI. In doing so, they can compare differences between these groups regarding the motor control strategies.

Response: Revised (Line 20-23 and line 106-109). The revised purpose is ‘‘Therefore, the purpose of this study was to compare the difference of the motor control strategy in ipsilateral and contralateral proximal joints while unperturbed walking and perturbed walking between individuals with CAI and matched controls.‘‘

3. Lines: 131-132: what questionnaires did the participants complete?

Response: Participants completed two questionnaires: (1) the Cumberland Ankle Instability Tool, and Foot and Ankle Ability Measure. The names of questionnaires were added to the manuscript (Line 139, 144-145). 

4. Lines 166-170: a graphic would assist the reader in determining how MVIC were performed or the authors can specify how the participants were positioned while performing the MVIC. It is also not clear what value from the MVIC was used to normalize the EMG

Response: (1) A graphic has been added in the manuscript (Line 182-190 and Fig 1). (2) The maximum value from three MVIC tests was used to normalize EMG. The sentence is revised as „The highest value (root mean squares, RMS) from three MIVC tests were applied to normalize the EMG data to each recorded muscle.“ (Line 214-216)

5. After reading the Data Processing within the Methods section, the authors stated that they compared average EMG profiles from 10 strides in each of the walking conditions. It is not clear how EMG and the perturbation were synchronized in order to accomplish this. The authors stated that a load cell was instrumented on the treadmill. Were these synchronized so that heel contact could be determined?

Response: The signals from acceleration sensors were synchronized with the EMG signals. The acceleration sensors were applied to detect heel contact and the onset of perturbation. The descriptions (Line 207-209 and line 220-223) and two figures (Fig 2 and Fig 3, Line 225-240) were added to the manuscript.

6. In the Results section, it is suggest ed that the sections titled “The magnitude of muscle activation “ and “The onset of muscle activation” be compartmentalized into analyses based upon the conditions (walking and perturbation). This will provide better clarity to the reader and better organization on the authors’ part. Similarly, tables 1 and 2 can be clarified further to match the text.

Response: The text of results and tables has been revised based on the reviewer's suggestion. (page 16-24, tables 1 and 2)

7. The authors indicated that other studies required participants to perform walking trials at 1.3-1.5 m/s, while the participants in this study walked at 1.0 m/s. It is not clear why this discrepancy was present, and this could influence the interpretation of the data with the literature.

Response: The corresponding discussion has been revised. (line 399-406)

Minor comments:

1. Lines 53-55: neuromuscular of ankle joint? Do the authors intend to state the neuromuscular control of the ankle joint?

Response: Yes, ‘‘state the neuromuscular control of the ankle joint‘‘ is what the authors mean. Revised (Line 61-62).

2. Lines 72-75: this sentence needs further clarification. Do the authors intend to state that the uninjured ankle is incapable of movement control that would reduce the potential for further injury to the already injured ankle?

Response: Yes, the manuscript was revised (Line 79-84).

3. Lines 87-89: “The compensation of external unexpected perturbations” – are the authors suggesting that compensatory mechanisms are essential to maintain system stability when external perturbations are presented? If so, this is not how their sentence is communicated

Response: The sentence is revised as: The ability to return to postural stability following external unexpected perturbations is important because it is the performance of core stability and relates to the risk of lower limb injuries (Line 96-98).

4. Lines 175-179: were these 10 to tal perturbations or 10 on the left and 10 on the right for a total of 20?

Response: Yes, revised. ‚for a total of 20 perturbations‘ was added into the manuscript. (Line 196-198).

5. Lines 180-184: was this same process used in the right side perturbation?

Response: The description has been revised. Perturbations in both sides were triggered by right heel contact but were programmed differently. The trigger of perturbation for the right side was triggered directly by right heel contact. The left side of perturbation was also triggered by right heel contact. The left side of perturbations occurred 200 ms after right heel contact plus stride duration (Line 198-205).

6. Lines 271-273: it is not clear what the authors are stating. Gmed had lowe r activation overall, but activated earlier at heel contact in the CAI group compared to controls?

Response: Revised as “Gmed overall had lower activation in the CAI group than controls. In the non-injured side, the CAI group activated Gmed earlier while heel contact than CON.“ (Line 334-335)

7. Lines 288-291: the shift of the center of mass t hat the authors bring forward is unclear. During the gait cycle the center of mass shifts laterally depending upon the limb supporting the body. Do the authors indicate that excessive lateral shift occurs in individuals with CAI?

Response: Yes, revised as ‘‘Gmed overall had lower activation in the CAI group than controls. In the non-injured side, the CAI group activated Gmed earlier while heel contact than CON‘‘. (Line 350-356)

8. Lines 315-317: if there is no st atistically significant difference, then there is no difference. Other areas of the Discussion section are stated similarly, and need to be adjusted accordingly (ex, lines 321-324). This section needs to be rewritten to reassess the authors’ findings with the current literature

Response: Revised. (Line 357-359, 378-379, 407-409)

9. Tables 1 and 2: are the statistics provided the comparisons between the healthy group and the CAI group? This is unclear in the legends

Response: The legends of tables 1 and 2 were revised. (Table 1: Group means, Cohen’s d effect size, and group difference for the magnitude and onset of muscle activation in the unperturbed walking trail; and Table 2: Group means, Cohen’s d effect size, and group difference for the magnitude and onset of muscle activation in the perturbed walking trail.)

If you require any additional information regarding our manuscript, please do not hesitate to contact us. Thank you. 

Sincerely,

Chiao-I Lin

---

## [Decision Letter · Decision Letter 1]

10 Feb 2021

The effect of chronic ankle instability on muscle activations in lower extremities

PONE-D-20-26118R1

Dear Dr. Lin,

We’re pleased to inform you that your manuscript has been judged scientifically suitable for publication and will be formally accepted for publication once it meets all outstanding technical requirements.

Kind regards,

Yumeng Li

Academic Editor

PLOS ONE

Additional Editor Comments (optional):

The study examined motor control strategy of individuals with CAI. The study is well designed and paper is well written. The authors have addressed all concerns from the reviewer. The paper could be accepted.

Reviewers' comments:

Reviewer's Responses to Questions

**Comments to the Author**

1. If the authors have adequately addressed your comments raised in a previous round of review and you feel that this manuscript is now acceptable for publication, you may indicate that here to bypass the “Comments to the Author” section, enter your conflict of interest statement in the “Confidential to Editor” section, and submit your "Accept" recommendation.

Reviewer #1: All comments have been addressed

2. Is the manuscript technically sound, and do the data support the conclusions?

Reviewer #1: Yes

3. Has the statistical analysis been performed appropriately and rigorously? 

Reviewer #1: Yes

4. Have the authors made all data underlying the findings in their manuscript fully available?

Reviewer #1: Yes

5. Is the manuscript presented in an intelligible fashion and written in standard English?

Reviewer #1: Yes

6. Review Comments to the Author

Reviewer #1: The authors have improved their manuscript to effectively communicate their study, and have addressed the concerns of this reviewer.

7. PLOS authors have the option to publish the peer review history of their article (what does this mean?). If published, this will include your full peer review and any attached files.

Reviewer #1: No

---

## [Editor Report · Acceptance letter]

12 Feb 2021

PONE-D-20-26118R1 

The effect of chronic ankle instability on muscle activations in lower extremities 

Dear Dr. Lin:

I'm pleased to inform you that your manuscript has been deemed suitable for publication in PLOS ONE. Congratulations! Your manuscript is now with our production department. 

Kind regards, 

on behalf of

Dr. Yumeng Li 

Academic Editor

PLOS ONE